# ProLonged Liposomal Delivery of TLR7/8 Agonist for Enhanced Cancer Vaccine

**DOI:** 10.3390/vaccines11091503

**Published:** 2023-09-19

**Authors:** Sehui Kim, Yeji Park, Jeonghun Kim, Sohyun Kim, Kyungmin Choi, Taegyun Kang, Inho Lee, Yong Taik Lim, Soong Ho Um, Chul Kim

**Affiliations:** 1Progeneer, 12 Digital-ro 31-gil, Guro-gu, Seoul 08380, Republic of Korea; shkim@progeneer.com (S.K.); yjpark@progeneer.com (Y.P.); jhkim@progeneer.com (J.K.); shkim1@progeneer.com (S.K.); kmchoi@progeneer.com (K.C.); tgkang@progeneer.com (T.K.); ihlee@progeneer.com (I.L.); sh.um@progeneer.com (S.H.U.); 2SKKU Advanced Institute of NanoTechnology (SAINT), Sungkyunkwan University (SKKU), Suwon 16419, Gyeonggi-do, Republic of Korea; yongtaik@skku.edu; 3School of Chemical Engineering, Sungkyunkwan University (SKKU), Suwon 16419, Gyeonggi-do, Republic of Korea; 4Department of Nano Science and Technology, Sungkyunkwan University (SKKU), Suwon 16419, Gyeonggi-do, Republic of Korea; 5Department of Nano Engineering, Sungkyunkwan University (SKKU), Suwon 16419, Gyeonggi-do, Republic of Korea; 6Biomedical Institute for Convergence at SKKU (BICS), Sungkyunkwan University, Suwon 16419, Gyeonggi-do, Republic of Korea; 7Institute of Quantum Biophysics (IQB), Sungkyunkwan University, Suwon 16419, Gyeonggi-do, Republic of Korea

**Keywords:** cancer vaccine, adjuvant, nanovaccines, TLR 7/8 agonist, melanoma, immunomodulator

## Abstract

Despite numerous studies on cancer treatment, cancer remains a challenging disease to cure, even after decades of research. In recent years, the cancer vaccine has emerged as a promising approach for cancer treatment, offering few unexpected side effects compared to existing therapies. However, the cancer vaccine faces obstacles to commercialization due to its low efficacy. Particularly, the Toll-like receptor (TLR) adjuvant system, specifically the TLR 7/8 agonist, has shown potential for activating Th1 immunity, which stimulates both innate and adaptive immune responses through T cells. In this study, we developed ProLNG-S, a cholesterol-conjugated form of resiquimod (R848), to enhance immune efficacy by stimulating the immune system and reducing toxicity. ProLNG-S was formulated as ProLNG-001, a positively charged liposome, and co-administered with ovalbumin (OVA) protein in the B16-OVA model. ProLNG-001 effectively targeted secondary lymphoid organs, resulting in a robust systemic anti-tumor immune response and tumor-specific T cell activation. Consequently, ProLNG-001 demonstrated potential for preventing tumor progression and improving survival compared to AS01 by enhancing anti-tumor immunity.

## 1. Introduction

Cancer, a complex and life-threatening disease, continues to be a considerable global health burden, impacting millions of lives annually [1,2]. Despite considerable advancements in conventional therapies such as surgery, chemotherapy, and radiation therapy, their efficacy is limited by systemic toxicity, acquired resistance, and adverse side effects [3]. In recent years, cancer vaccine has emerged as a promising and rapidly evolving field, utilizing the immune system to selectively target and eliminate cancer cells, surpassing traditional treatments [4,5].

Among the various immunotherapeutic approaches, Toll-like receptor (TLR) agonists have garnered considerable attention due to their ability to stimulate potent immune responses [6,7,8,9,10,11]. Consequently, some TLR agonists, including MPLA and CpG, have been approved for commercial use as adjuvants [12]. Imiquimod and resiquimod (R848), an imidazoquinolin-like molecule that binds to TLR7/8 [4], has shown promise as an immunomodulator for skin pathologies and cancers [13]. It enhances T cell proliferation, activation, and immune cell infiltration into the tumor microenvironment. Notably, TLR 7/8 agonists (TLR7/8a) have gained attention for their potential to augment the effectiveness of cancer immunotherapy by activating CD8+ T cells [14]. CD8+ T cells, also known as cytotoxic T lymphocytes, play a crucial role in immune responses [15,16,17] against cancer. These specialized immune cells can recognize and destroy cancer cells through the identification of tumor-specific antigens presented on the malignant cell surface [15,18,19]. Once activated, CD8+ T cells undergo clonal expansion, leading to a potent cytotoxic immune response that selectively targets and eliminates cancer cells throughout the body [1,19].

However, despite the anti-tumor efficacy of TLR 7/8a, the clinical applicability of TLR7/8a in cancer immunotherapy is currently limited to skin cancers, including metastatic cancers that present on the skin, due to severe systemic toxicity upon distribution [20]. These toxicities include lymphopenia, anemia, and flu-like symptoms.

In this study, we have conjugated the active site of R848 with cholesterol through a disulfide bond, resulting in a compound named ProLNG-S (Appendix A) [21,22]. The disulfide bond of ProLNG-S can be cleaved by enzymes or physiochemical conditions within cells, leading to prolonged activation of immune responses and amplification of T cell responses. ProLNG-S is selectively activated in the endosomal environment, exhibiting low toxicity while enhancing T cell-mediated immune efficacy with Th1 polarization [22]. To maximize cellular uptake, we have developed a cationic liposomal carrier for ProLNG-S, named ProLNG-001. ProLNG-001 possesses the optimal size and surface charge for facilitating cellular uptake. In this paper, we demonstrate the enhanced overall immune efficacy and reduced toxicity of ProLNG-001. Specifically, we provide evidence of its anti-tumor efficacy through the activation of T cells and natural killer (NK) cells [23,24,25,26,27] in the B16-ovalbumin (OVA) cancer model, comparing its effectiveness to AS01, an approved commercial adjuvant.

## 2. Materials and Methods

### 2.1. Synthesis of ProLNG-001 (Liposomes Containing ProLNG-S) and Blank Liposome

ProLNG-001 was synthesized by using a microfluidics method with two different phases of solutions. The lipid phase was prepared by mixing ethanol and dimethyl sulfoxide (DMSO) (Sigma-Aldrich, St. Louis, MO, USA) in a volume ratio of 9:1. The water phase was prepared using tris-buffered saline (TBS) at pH 7.4 (WELGENE, Gyeongsan, Republic of Korea). To prepare ProLNG-001, 1,2-dioleoyl-sn-glycero-3-phosphocholine (DOPC, Avanti Polar Lipids, Alabaster, AL, USA), 1,2-dioleoyl-3-trimethylammonium-propane (DOTAP, Avanti Polar Lipids), and ProLNG-S were dissolved in the lipid phase at a weight ratio of 2:2:1. After preparing the lipid-containing solution, the two solutions were combined in a volume ratio of 3:1 at a flow rate of 12 mL/min using the NanoAssemblr^®^ Ignite (Precision NanoSystem, Vancouver, BC, Canada). Additionally, the concentration of ProLNG-001 was calculated by absorbance data from UV-vis spectrometry [22]. Subsequently, the solution was filtered using a 0.8 μm syringe filter (Minisart^®^, Sartorius, Göttingen, Germany) and dialyzed with TBS overnight to remove organic solvents. Blank liposome was synthesized by the described method above by substituting the same mole number of cholesterol (Sigma-Aldrich) for ProLNG-S.

### 2.2. Physicochemical Properties of ProLNG-001

The size and morphology of ProLNG-001 were analyzed using dynamic light scattering (DLS) (LiteSizer, Anton-Paar, Graz, Austria) and cryo-transmission electron microscopy (cryo-TEM). For DLS analysis, 100 μL of ProLNG-001 was diluted in TBS and measured. For cryo-TEM, 3 μL of the sample were applied to glow-discharged Quantifoil R1.2/1.3 Cu 300 grids and flash frozen in liquid ethane using the Vitrobot Mark IV (Thermo Fisher Scientific, Waltham, MA, USA) set at 100% humidity and 4 °C for the preparation chamber, with a 3–5 s blot time. Cryo-TEM micrographs were obtained using the Glacios microscope (Thermo Fisher Scientific), operated at an accelerating voltage of 200 kV with a 70 μm C2 aperture. Images of the samples were acquired using a Falcon III direct electron detector in linear mode with a 100 μm objective aperture.

### 2.3. Activation of Bone Marrow-Derived Dendritic Cells (BMDCs)

BMDCs were generated from the bone marrow of female 6- to 8-week-old C57BL/6 mice (Orient Bio, Seongnam, Republic of Korea). BMDCs were differentiated and seeded according to a previous study [28]. BMDCs were treated with ProLNG-001 (1 μg/mL) or the equivalent amount of R848 (0.5 μg/mL) for 24 h. To measure cytokine levels, culture supernatants were collected, and interleukin (IL)-12, IL-10, and tumor necrosis factor (TNF)-α secretions were measured using the OptEIA enzyme-linked immunosorbent assay (ELISA) kit (BD Biosciences, San Jose, CA, USA) following the manufacturer’s instructions. Absorbance at a wavelength of 450 nm was measured using the SpectraMax M5e multi-mode microplate reader (Molecular Devices, San Jose, CA, USA) to quantify cytokine concentrations. To confirm the maturation of BMDCs, fluorescence-activated cell sorting (FACS) analysis was performed. Harvested cells were washed with phosphate-buffered saline (PBS, Hyclone, Cytiva Life Sciences, Logan, UT, USA) and stained with APC-CD11b, FITC-CD11c, PerCP-CD86, and APC-Cy7-Live/Dead antibodies following the standard sampling method provided by the antibody manufacturer. The stained cells were measured using the Sony MA900 flow cytometer (Sony Biotechnology, San Jose, CA, USA).

### 2.4. Cytotoxic T-Lymphocyte Response in Human Peripheral Blood Mononuclear Cells (hPBMCs)

hPBMCs (Lonza, Basel, Switzerland) were cultured in a lymphocyte growth medium (LGM-3, Lonza) supplemented with 10% heat-inactivated fetal bovine serum (FBS, Thermo Fisher Scientific) and 1% penicillin-streptomycin (Thermo Fisher Scientific). Cells were seeded at a density of 1 million cells per well in a 24-well cell culture plate (SPL Life Sciences, Pocheon, Republic of Korea) and treated with ProLNG-001 at the indicated concentration. For cytokine measurement, cells were incubated for 24 h at 37 °C in a humidified 5% CO_2_ incubator. After 24 h, the cell culture supernatants were collected, and the IFN-γ secretion was measured using the IFN-gamma Human Uncoated ELISA Kit (Thermo Fisher Scientific). For the analysis of IFN-γ-secreting CD8+ T cells, FACS analysis was performed. The cells were stimulated with a cell stimulation cocktail (plus protein transport inhibitors 0.5×, Thermo Fisher Scientific) for 6 h, after 48 h of incubation with the samples. After stimulation, the cells were intracellularly stained using the Intracellular Fixation & Permeabilization Buffer Set Kit (Thermo Fisher Scientific) following the manufacturer’s recommendations. The cells were stained with APC-Cy7-Live/Dead (Thermo Fisher Scientific), BV421-CD3, PE-CD8, and PerCP-Cy5.5-IFN-γ antibodies (BioLegend, San Diego, CA, USA).

### 2.5. Measurement of IgG Titer in Mouse Plasma

To determine the titer of IgG subclasses in mouse plasma, female specific-pathogen-free C57BL/6 mice were obtained from Orient Bio at 6 weeks of age and used for experiments after one week. The mice were subcutaneously vaccinated with a mixture of OVA (albumin from chicken egg white, grade V, Sigma-Aldrich) and ProLNG-001 twice with 2-week intervals on the upper right back. The mice plasma was collected two weeks after the last vaccination. The plasma was separated from the blood by centrifugation at 13,000× *g* rpm for 30 min, and the quantity of specific antibodies against the OVA protein in the mouse plasma was measured by ELISA. For this purpose, 2 μg/mL of OVA protein was coated onto a 96-well immune plate overnight. The plates were then blocked with 5% (*w*/*v*) skim milk (BD Bioscience) at 37 °C for 1 h. Gradient-diluted mouse plasma was added to the wells and incubated for 2 h. Horseradish-peroxidase (HRP)-conjugated goat anti-mouse IgG (H+L) (SouthernBiotech, Birmingham, AL, USA), HRP-conjugated rat anti-mouse IgG1 (SouthernBiotech), and HRP-conjugated goat anti-mouse IgG2c (SouthernBiotech) were used as secondary antibodies, respectively. Subsequently, an indirect ELISA was performed using 3,3′,5,5′-tetramethylbenzidine (TMB, BD Biosciences) as the substrate. Color development was stopped by adding 2M H_2_SO_4_, and the plates were read at 450 nm using a SpectraMax M5e plate reader (Molecular Devices) according to the manufacturer’s recommendations.

### 2.6. Analysis of Delayed Cytokine Response in Mouse Plasma

To confirm the delayed cytokine expression with ProLNG-001, the level of tumor necrosis factor (TNF)-α in mouse plasma was measured. C57BL/6 mice were subcutaneously administered TBS, R848 (25 μg), or ProLNG-001 (140 μg). Mouse plasma was collected at various time points (0, 1, 3, 6, and 24 h) following administration. The plasma was separated from heparinized blood by centrifugation at 13,000× *g* rpm for 30 min, and the level of TNF-α in the isolated mouse plasma was analyzed using the BD OptEIA Mouse TNF (Mono/Mono) ELISA Set (BD Biosciences) following the manufacturer’s instructions.

### 2.7. Hematoxylin and Eosin Staining

Mouse liver samples were dissected, mounted in an OCT embedding compound, and frozen at −20 to −80 °C. A Cryostat was used to cut 5 µm thick tissue sections. After air drying, the frozen sections were fixed in 10% neutral buffered formalin (Abelbio, Seoul, Republic of Korea) for 10 min and washed with water. The slides were then rinsed in 1× PBS twice for 3 min each to remove OCT or other tissue-embedding compounds. A gentle stream of tap water was used to rinse the slides for 5 min. Mayer’s Hematoxylin (Sigma-Aldrich) was used to stain the sections for 90 s, followed by a gentle stream of tap water wash for 1 min. The sections were dipped in 70% ethanol (Millipore, Burlington, MA, USA) for 30 s, followed by a dip in 95% ethanol for 30 s. Alcoholic-Eosin (Sigma-Aldrich) was used for counterstaining for 30 s. Dehydration was carried out by passing the sections through 2 changes of 95% ethanol for 15 s, followed by three changes of 100% ethanol for 15 s. The sections were then cleared through three changes of xylene (Sigma-Aldrich) for 1 min each. Finally, the slides were mounted with a mounting solution (Sigma-Aldrich) using glass coverslips (Marienfeld, Germany). The slide was scanned by Pannoramic 250 Flash 3 (3D HISTECH, Budapest, Hungary) with 100× magnification (10× optical magnification, 10× objective magnification). 

### 2.8. Assessment of the Anti-Cancer Therapeutic Effect of ProLNG-001

To evaluate the therapeutic effect of the ProLNG-001 adjuvanted cancer vaccine, the B16-OVA mouse melanoma cell line (Merck Millipore) expressing ovalbumin was cultured according to previously described methods. Tumors were established on the upper right back of C57BL/6 mice by subcutaneously injecting 3 × 10^5^ B16-OVA cells. Vaccinations were administered as indicated in Table 1 twice and Table 2 three times once a week starting on the 3rd day after tumor implantation. To evaluate the efficacy of each group, tumor volume was calculated using an electronic caliper at intervals of 3 to 4 days using the equation: tumor volume (mm^3^) = 0.5 × length × (width)^2^. Mice were sacrificed when the tumor volume reached 2000 mm^3^. To analyze the immune cell recruitment in tumors and spleens, mice were sacrificed 7 days after the last immunization, and tumors and spleens were isolated. The collected tissues were chopped and resuspended in collagenase I (100 units/mL; Thermo Fisher Scientific) and collagenase V (100 units/mL; Thermo Fisher Scientific) at 37 °C for 30 min. The homogenized samples were passed through a 70 μm cell strainer (Falcon, Glendale, AZ, USA) to obtain a single-cell suspension, and to perform red blood cell lysis (BioLegend). The cells were washed with fresh PBS twice, stained with FITC-CD4, APC–CD3, PE-CD8, BV605–IFN-γ, PerCP-Cy5.5–TNF-α, FITC-NK1.1, and BV421-CD69–antibodies (BioLegend), and analyzed using flow cytometry. 

For the analysis of antigen-specific CD8+ T cells, single cells (5 × 10^5^ per well) from the spleen and tumor were stimulated for 18 h with 10 μg of the SIINFEKL peptide (Mimotopes, Mulgrave, Victoria, Australia) and a stimulation cocktail, as described previously (Section 2.3).

### 2.9. Statistical Analysis

Student’s *t*-test and one-way analysis of variance were employed for the statistical analyses, and significance was denoted by *p* < 0.05. All values are presented as means ± standard deviations. The survival rates were determined using the Mantel-Cox chi-square test. GraphPad Prism was utilized for all the statistical analyses.

## 3. Results

### 3.1. Synthesis and Characterization of ProLNG-001

To evaluate the therapeutic potential and the ease of manufacturing our novel adjuvant, we developed a liposomal formulation incorporating ProLNG-S. Given the hydrophobic nature of ProLNG-S, which contains a cholesterol moiety, we successfully incorporated it between the lipid layers of liposomes composed of DOPC and DOTAP. To confirm the synthesis of liposomes, we assessed the size, surface charge, and encapsulation efficiency of ProLNG-S in ProLNG-001 using DLS and UV-Vis spectrometry.

ProLNG-001 exhibited uniform synthesis with a narrow size distribution, measuring an average size of 83.0 ± 1.2 nm (Figure 1a). The positive charge of ProLNG-001 was attributed to the influence of DOTAP, resulting in a charge of 29.9 ± 1.7 mV (Figure 1b). Cryo-TEM further confirmed the uniform layered structure and spherical shape of ProLNG-001 (Figure 1c). 

### 3.2. Cytokine Secretion and Activation Marker Expression of Dendritic Cells Induced by ProLNG-001 in Mouse BMDCs

To investigate the effect of ProLNG-001 on the proliferation and maturation of dendritic cells (DCs), we conducted MTS assays, ELISA, and flow cytometry analyses using BMDCs treated with various samples and incubated for 24 h. Appendix A demonstrates that as the concentration of ProLNG-001 increased, BMDC proliferation exhibited a proportional response. Furthermore, we performed an ELISA assay to assess the secretion of IL-12 and TNF-α, a cytokine crucial for Th1 immunity [29,30]. The IL-12 and TNF-α expressed only in the group of BMDCs treated with ProLNG-001 (Figure 2a). Additionally, we showed the Th1 polarization effect of ProLNG-001 and compared it with R848 in BMDCs by the ratio of IL-12 to IL-10 (Appendix A). The production of IFN-α, a member of the type 1 interferon family, was also quantified in pDC using ELISA, alongside ProLNG-001 and R848. Notably, ProLNG-001 exhibited an eightfold higher secretion of IFN-α compared to an equivalent amount of R848 (Appendix A).

Flow cytometry analysis revealed an increased frequency of co-stimulatory markers (CD80, CD86) expression on CD11b+CD11c+ DCs in the ProLNG-001 treated group (Figure 2b). The percentages of CD80+ and CD86+ expression were approximately 66%, respectively, and were higher in ProLNG-001-treated BMDCs compared to the controls. Blank liposomes exhibited similar levels of co-stimulatory marker expression to TBS-treated BMDCs. The percentage of CD80+CD86+ expression was significantly increased up to 61% in ProLNG-001-treated BMDCs, which is 1.8 times compared to blank liposomes. Importantly, ProLNG-001 promoted a higher population and induced maturation effectively without severe toxicity in BMDCs.

### 3.3. IFN-γ Secreting T Cells in Human Peripheral Blood Mononuclear Cells (hPBMCs)

To evaluate the impact of ProLNG-001 on T cell immunity, the population of IFN-γ-secreting T cell subsets (CD3+CD4+, CD3+CD8+) was measured by using flow cytometry. Additionally, the total quantity of secreted IFN-γ in hPBMCs was assessed by ELISA. Figure 3a demonstrates a significant increase in the frequency of CD8+IFN-γ+ and CD4+IFN-γ+ T cells within the CD3+ T cell population upon treatment with ProLNG-001 compared to the control group. Notably, the proportion of CD8+IFN-γ+ cells was three-fold higher than the control. Furthermore, hPBMCs were treated with various concentrations of ProLNG-001 for 24 h, and IFN-γ production was measured. The results revealed a dose-dependent increase in IFN-γ production in response to ProLNG-001, with higher concentrations yielding greater amounts of IFN-γ secretion (Figure 3b).

### 3.4. Delayed Cytokine Release in Mouse Plasma Induced by ProLNG-001

To assess the potential toxicity of ProLNG-001, we performed subcutaneous injections of TBS, R848, and ProLNG-001 in C57BL/6 mice. We measured TNF-α levels in mouse plasma using ELISA (Appendix A). The R848 group exhibited a rapid increase in TNF-α concentration within an hour, which declined to basal levels after 3 h. In contrast, the ProLNG-001 group showed a lower maximum TNF-α concentration in plasma compared to the R848 group, and the release of TNF-α was sustained beyond 3 h post-injection. While R848 induced rapid cytokine secretion, ProLNG-001 exhibited a sustained and lowered release of cytokines.

To further investigate the potential toxicity of ProLNG-001, we monitored body weight changes and evaluated plasma toxicity [31,32]. C57BL/6 mice received subcutaneous injections of TBS, ProLNG-001 (140 μg), or ProLNG-001 × 4 (560 μg) four times at weekly intervals, followed by a 2-week recovery period. The ProLNG-001-treated mice did not show significant body weight loss compared to the control group (Figure 4a). Additionally, a comprehensive toxicity assessment of plasma samples from each group revealed no significant differences compared to the control group (Appendix A) [33,34]. Furthermore, hepatic histopathological evaluation was performed through H&E staining for all groups [35]. Mice treated with ProLNG-001 did not exhibit any damage or histological changes, and liver toxicity was not observed anywhere after administration (Figure 4b).

To analyze the Th1-skewed immune response induced by ProLNG-001, we have incorporated data on IgG subclasses [36,37,38]. Overall, when ProLNG-001 was administered with the OVA antigen, total IgG production increased compared to the antigen-only group. Additionally, IgG2c was only detected in the plasma of mice immunized with ProLNG-001 and the antigen, with no IgG2c present in the OVA-immunized group (Appendix A). Based on the data showing changes in IgG subclasses, ProLNG-001 induced robust antibody production but also promoted strong Th1 immunity, which is essential for its anti-tumor effects [39].

### 3.5. Anti-Tumor Effects and Recruitment of Activated Immune Cells by ProLNG-001

We investigated the anti-tumor effects of ProLNG-001 in the B16-OVA melanoma-bearing mouse model, focusing on immune cell responses. The final dose of ProLNG-001 in this experiment was decided according to the preliminary experiment (Appendix A). Subcutaneous inoculation of B16-OVA melanoma cells on the upper right back of C57BL/6 mice was followed by two immunizations, administered at 1-week intervals, with each sample described in Figure 5. The group treated with OVA protein alone did not exhibit tumor reduction compared to the control (TBS) group, and there were no significant differences between these groups. However, the ProLNG-001/OVA group showed suppressed tumor growth, with tumors reaching an average size one-third that of the other groups by day 17 (Figure 5a).

To evaluate the anti-tumor effects of ProLNG-001, we analyzed the spleen, a secondary lymphoid organ, and the tumor microenvironment (TME). Firstly, we observed an increase in the number of T cells in the spleens of the ProLNG-001 group compared to the other groups. Specifically, the number of both CD4+ T cells (CD3+CD4+) and CD8+ T cells (CD3+CD8+) were elevated (Appendix A). Moreover, antigen-specific T cells secreting cytokines, such as TNF-α and IFN-γ, were quantified using flow cytometry on harvested splenocytes. These splenocytes were re-stimulated with OVA protein and a cell stimulation cocktail for 18 h, followed by intracellular staining. ProLNG-001 significantly enhanced the frequency of antigen-specific IFN-γ-secreting CD8+ T cells in response to antigen stimulation (Figure 5b). IFN-γ is a pleiotropic cytokine known for its anti-tumor function and is mainly regulated by CD8+ and CD4+ T cells during adaptive immune responses. Notably, in the TME, IFN-γ can act as a cytotoxic cytokine, along with perforin and granzyme B, inducing apoptosis in tumor cells [40,41]. Additionally, ProLNG-001 prominently increased the population of TNF-α-secreting effector CD8+ T cells compared to the other groups (Figure 5c). TNF-α is an important cytokine produced by activated NK cells, macrophages, and T lymphocytes, and it exhibits cytotoxic effects on tumors [42]. The ProLNG-001 group displayed approximately double the percentage of IFN-γ and TNF-α-secreting CD8+ T cells compared to the antigen-only group, indicating a higher ratio of antigen-specific effector T cells secreting anti-tumor cytokines in the spleen, which may be associated with tumor growth regression (Figure 5d, Appendix A).

Next, we examined immune cells in the TME. We measured the proportion of tumor-infiltrating T cells and NK cells, as well as the expression of activation markers. ProLNG-001 significantly increased the infiltration of T cells (CD3+) into the tumor, surpassing a two-fold increase compared to the other groups (Figure 5e). ProLNG-001 also tended to enhance the number of NK cells (CD3-NK1.1+) infiltrating the tumor (Figure 5f). Particularly noteworthy points were the differences in activation marker expression among tumor-infiltrating T cells and NK cells. CD69 is a classical early marker of lymphocyte activation, which was rapidly expressed on the surface of T lymphocytes upon TCR/CD3 engagement [43]. CD69 expression is known to play an important role in regulating the severity of various murine inflammation models, including tumor immunity [44]. The expression of the activation marker CD69 on tumor-infiltrating T cells and NK cells significantly increased following immunization with ProLNG-001 (Appendix A). Specifically, ProLNG-001 exhibited approximately three times higher levels of activated T cells and activated NK cells compared to the antigen-only group (Figure 5g,h). These findings indicate that ProLNG-001 enhances the number of antigen-specific effector T cells in the spleen, as well as tumor-infiltrating activated T cells and NK cells with anti-tumor effects.

To provide evidence of the anti-tumor effects mediated by antigen-specific T cells, a further experiment was conducted in a prophylactic model. When ProLNG-001 was administered with the antigen, all individuals were completely protected from tumor growth. On the other hand, the ProLNG-001 group without the antigen exhibited prophylactic effects with similar efficacy to the antigen-only group. (Appendix A). This data emphasized that ProLNG-001 enhanced not only innate immune response but also antigen-specific T cells strongly. 

### 3.6. Enhanced Anti-Tumor Effects of ProLNG-001 Compared to Commercial Adjuvants

In the subsequent investigation, we compared the anti-tumor effects and survival outcomes of ProLNG-001 with those of other TLR vaccine adjuvants. B16-OVA melanoma-bearing mice were immunized with a combination of antigen and TLR adjuvants. The samples were administered subcutaneously three times at weekly intervals, starting on day 3 after tumor inoculation, and tumor size was monitored for 24 days. Mice injected with antigen alone exhibited rapid tumor growth, and the survival rate began to decline after 2 weeks of tumor inoculation. In contrast, immunization with the combination of antigen and ProLNG-001 or AS01 showed suppressive effects on tumor growth compared to the other groups (Figure 6a). TLR adjuvants delayed tumor growth and improved survival rates. Particularly when an antigen was administered with ProLNG-001, the survival rate remained 100% until day 24, while the control and antigen-only groups reached 0%, and the AS01/OVA group exhibited a survival rate of 40% (Figure 6b). Importantly, the ProLNG-001 vaccination not only delayed tumor growth but also significantly improved overall survival.

## 4. Discussion

The present study on cancer vaccine therapy underscores the importance of adjuvants in combating the immune-suppressive tumor microenvironment. Activation of TLR7/8 receptors in immune cells triggers downstream signaling pathways involving MyD88 which in turn induces the activation of NF-κB and IRFs [45,46,47,48]. It ultimately leads to the production of pro-inflammatory cytokines and the activation of antigen-presenting cells [49,50,51]. ProLNG-001 has been specifically designed to optimize its efficacy by modulating its particle characteristics, thereby inducing an immune response without toxicity. The size of ProLNG-001 has been optimized for effective targeting of immune cells while circulating in the lymphatic system [52]. Additionally, the positive charge of ProLNG-001 facilitates cellular uptake by enabling penetration of the cell membrane, which is particularly relevant since immune cells possess a negative surface charge ranging from −40 mV to −80 mV [53]. Leveraging these unique attributes of ProLNG-001 fosters tumor regression by stimulating Th1-polarized cytotoxic T cells and tumor-infiltrated NK cells through the immune response mediated by TLR7/8 receptors.

Results obtained from BMDCs and pDCs demonstrate that ProLNG-001 induces superior dendritic cell maturation, IL-12 cytokine secretion, and IFN-α secretion. By utilizing ProLNG-001, we mitigate the toxic effects associated with R848, as revealed by previous studies [22], while ensuring sustained cytokine secretion. The change in IgG subclasses showed a Th1-skewed immune response induced by ProLNG-001, as IgG2c was only detected in the plasma of mice immunized with ProLNG-001 and the antigen. The observed increase in pro-inflammatory cytokines, such as IFN-γ and TNF-α, further supports the notion that ProLNG-001 fosters an immune stimulatory environment, thereby enhancing anti-tumor immune responses and promoting tumor regression. Moreover, our study highlights the activation of CD8+ T cells and NK cells within the tumor following ProLNG-001 administration. This finding suggests that the adjuvant effect of ProLNG-001 facilitates the activation of cytotoxic T cells, which play a crucial role in mediating the anti-tumor effect [54,55,56]. 

We applied ProLNG-001 at the concentration of TLR7/8a frequently used in clinical trials [57] converted to the murine doses in the B16-OVA model. A key finding of this study was the significant increase in survival observed in the ProLNG-001 group compared to the other groups. It also exhibited an excellent effect in terms of tumor reduction although it did not show a significant difference compared to AS01. These results suggest prolonged TLR7/8 stimulation by liposomal adjuvant combined with an antigen may benefit cytokine cascade activation after antigen delivery and therefore induction of more integrative immune protections. Despite the promising results, it is important to acknowledge several limitations. Firstly, the B16-OVA cancer model employed in this study represents a simplified system and may not fully capture the complexity of human tumors. Therefore, further investigations using clinically relevant models are warranted to validate the potential of ProLNG-001 in cancer vaccine therapy. Additionally, the long-term effects of ProLNG-001 were not extensively evaluated in this study. Future research should prioritize long-term analyses, including re-challenge experiments, and comprehensively assess any potential adverse effects associated with ProLNG-001 administration to ensure its clinical applicability.

## 5. Conclusions

This study demonstrates that ProLNG-001 induces robust activation of T cells and NK cells, leading to a favorable impact on tumor regression in the B16-OVA model. These findings provide a strong rationale for further exploration of liposomal prolonged TLR7/8a as a promising adjuvant in cancer vaccine therapy. Future investigations should prioritize evaluating the efficacy and safety of ProLNG-001 in clinically relevant models, as well as optimizing its therapeutic potential in combination with other immunotherapeutic strategies.

## 6. Patents

Progeneer Inc. holds the patent for the subject matter discussed in this article, identified as PCT/KR2020001753, titled “TOLL-LIKE RECEPTOR 7 OR 8 AGONIST-CHOLESTEROL COMPLEX AND METHOD OF PREPARING SAME”. This patent has been successfully filed or granted in multiple jurisdictions, including KR, AU, BR, CA, CN, EP, IN, JP, MX, RU, and US.

## Figures and Tables

**Figure 1 vaccines-11-01503-f001:**
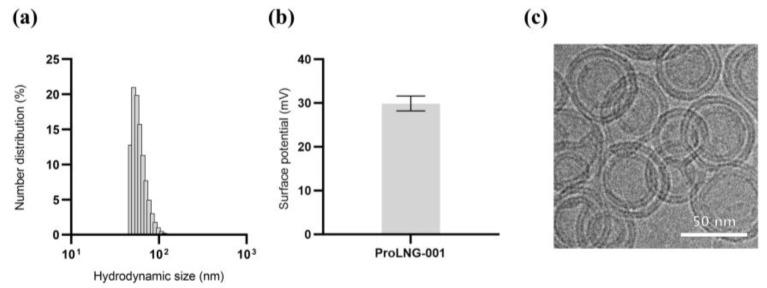
Particle characterization of ProLNG-001 liposomes containing ProLNG-S. (**a**) Number distribution and (**b**) surface potential of ProLNG-001 was measured by DLS. (**c**) Cryo-TEM image of ProLNG-001 demonstrates its uniform layered structure and spherical shape.

**Figure 2 vaccines-11-01503-f002:**
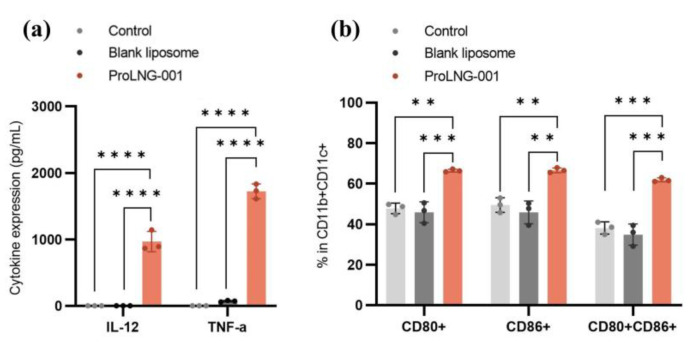
In-vitro immune activation and cytokine secretion of liposomes containing ProLNG-S (ProLNG-001). BMDCs were incubated with ProLNG-001 1 μg/mL for 24 h. (**a**) IL-12 (p70) and TNF-α production in BMDCs (*n* = 3). (**b**) The expression of maturation marker on BMDCs was assessed by flow cytometry (*n* = 3). ** *p* < 0.01, *** *p* < 0.001, **** *p* < 0.0001.

**Figure 3 vaccines-11-01503-f003:**
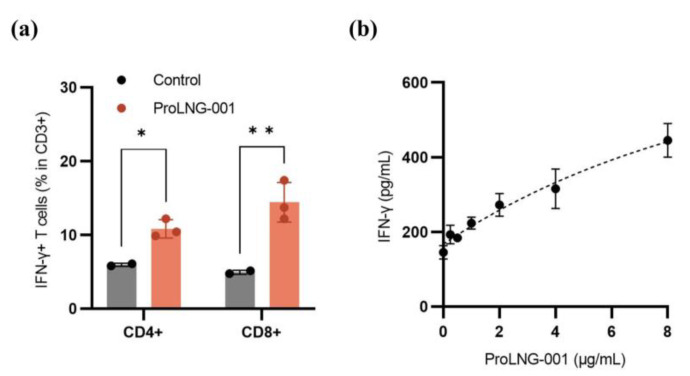
In-vitro immune profile of ProLNG-001 measured in hPBMCs. (**a**) After treatment with ProLNG-001 for 48 h, cells were re-stimulated with a cell stimulation cocktail for 6 h. The ratio of IFN-γ-secreting CD4+ T cells and CD8+ T cells among CD3+ T cells was determined by flow cytometry (*n* = 3). (**b**) IFN-γ production in hPBMCs following treatment with indicated concentrations (0.25, 0.5, 1, 2, 4, and 8 μg/mL) of ProLNG-001 for 24 h. Cytokine secretion was measured by ELISA after collecting supernatants (*n* = 3). * *p* < 0.05, ** *p* < 0.01.

**Figure 4 vaccines-11-01503-f004:**
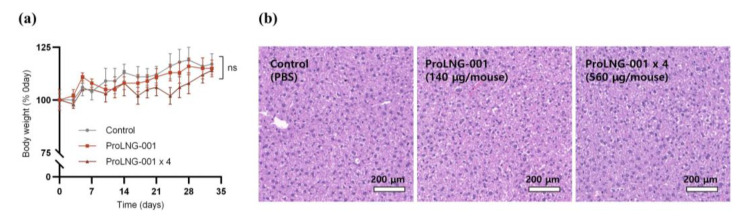
Evaluation of ProLNG-001 toxicity in mice. C57BL/6 mice received subcutaneous injections of TBS, ProLNG-001 (140 μg), or ProLNG-001 × 4 (560 μg) four times at weekly intervals. (**a**) Body weight was monitored every 3–4 days following a 2-week recovery period after the final injection (*n* = 5). (**b**) Representative histologic investigation of the liver with H&E staining after two weeks of recovery from the last injection (*n* = 5). ns: not significant.

**Figure 5 vaccines-11-01503-f005:**
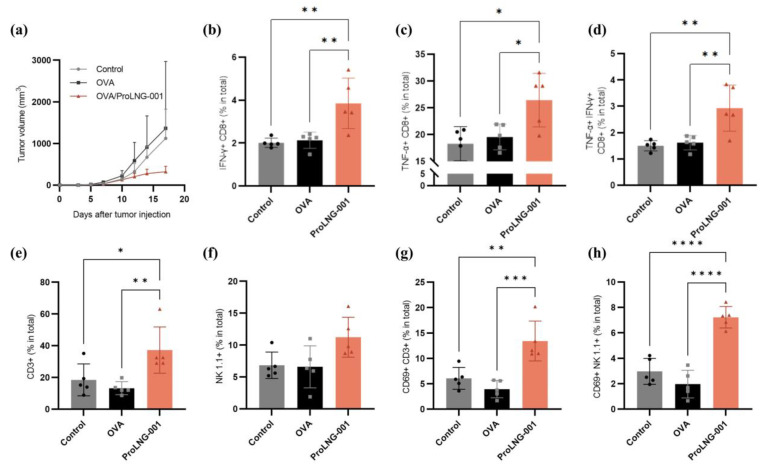
The anti-tumor immune response induced by ProLNG-001 in the B16-OVA tumor-bearing mouse model. C57BL/6 mice were subcutaneously injected with 3 × 10^5^ B16-OVA cells on the upper right back. On days 3 and 10 after tumor inoculation, subcutaneous injections of TBS, OVA (10 μg), or ProLNG-S (140 μg) + OVA (10 μg) were administered. Tumor growth was measured at least every two or three days. (**a**) Tumor growth curves are shown (*n* = 5/group). T cell analysis in splenocytes was performed using flow cytometry on day 17. To assess antigen-specific T cells, splenocytes were re-stimulated with 10 μg/mL OVA and a cell stimulation cocktail (0.5×) for 18 h. The percentage of (**b**) IFN-γ-secreting CD8+ T cells, (**c**) TNF-α-secreting CD8+ T cells, and (**d**) TNF-α+IFN-γ+CD8+ T cells are shown. T cell and NK cell activation in the TME was measured by flow cytometry on day 17. The percentage of (**e**) T cells (CD3+) and (**f**) NK cells (CD3-NK1.1+) among total tumor-infiltrating cells is presented. The percentage of (**g**) activated T cells (CD69+CD3+) and (**h**) activated NK cells (CD69+CD3-NK1.1+) are depicted. Dots in the bar graph represent the values of individual mice (*n* = 5/group). Error bars indicate means with standard deviations. * *p* < 0.05, ** *p* < 0.01, *** *p* < 0.001, **** *p* < 0.0001.

**Figure 6 vaccines-11-01503-f006:**
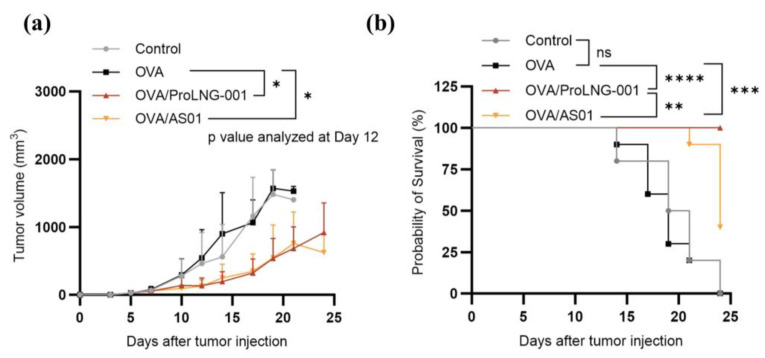
Tumor growth and survival analysis of B16-OVA tumor-bearing mice treated with antigen and various adjuvants. C57BL/6 mice were subcutaneously injected with 3 × 10^5^ B16-OVA cells on the upper right back. On days 3, 10, and 17 after tumor inoculation, subcutaneous injections of TBS, OVA (10 μg), ProLNG-001 (140 μg) + OVA (10 μg), and AS01 (5 μg) + OVA (10 μg) were administered. Tumor growth was monitored three times a week for 24 days. (**a**) Tumor growth. (**b**) The survival rate with Mantel-Cox chi-square test of B16-OVA tumor-bearing mice. Tumor volumes greater than 2000 mm^3^ were considered indicative of death (*n* = 10/group). *ns*: not significant, * *p* < 0.05, ** *p* < 0.01, *** *p* < 0.001, **** *p* < 0.0001.

**Table 1 vaccines-11-01503-t001:** Sample information for the analysis of anti-cancer therapeutics in B16-OVA. (*n* = 5).

No.	Sample Information
Group 1	TBS
Group 2	OVA 10 μg
Group 3	OVA 10 μg + ProLNG-001 140 μg

**Table 2 vaccines-11-01503-t002:** Sample information for the comparison with commercial adjuvants in B16-OVA. (*n* = 10).

No.	Sample Information
Group 1	TBS
Group 2	OVA 10 μg
Group 3	OVA 10 μg + ProLNG-001 140 μg
Group 4	OVA 10 μg + AS01 5 μg

## Data Availability

The datasets generated or analyzed during this study are available when the reasonable request is suggested.

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
