# Peer review of "ProLonged Liposomal Delivery of TLR7/8 Agonist for Enhanced Cancer Vaccine"

_vaccines, 2023, doi:10.3390/vaccines11091503_

Round 1

Reviewer 1 Report

The manuscript by Kim et al addresses the effect of TLR7/8 agonist ProLNG001 as adjuvant for cancer treatment. This manuscript is presenting interesting data however the draft has a number of points that need to be addressed:

  1. While the focus of the study is directed to Th1 immune responses through T cells, authors could include and investigate macrophage polarisation which plays significant role in Th1 responses and release of proinflammatory TLR-dependent mediators. I suggest the use of PMBC-derived macrophages.
  2. TLR/TRIF signalling pathways could be explored in the study by measurement of IFN beta/IP-10 in response ProLNG001.
  3. The focus of the study is comparative analysis between ProLNG001 and R848. Authors provided in vivo and in vitro data about toxicity of ProLNG001. I suggest authors to include control and R848 groups for in vitro measurement of cell viability and proliferation. In respect to in vivo data, measurement of liver enzymes, in addition to histological analysis could be beneficial.
  4. The authors should justify in detail the use of 1mg/ml (in vitro) and 140mg (in vivo) of ProLNG001.
  5. Importantly, this study demonstrated that ProLNG001 increased the survival rate in animal group treated with ProLNG001 compared to AS01 treated and control groups. However, this effect is visible up to 25 days. Additional information about long term effects could benefit the conclusions of this study.

.

Reviewer 2 Report

The manuscript by Sehui Kim et al., shows the effects of a cholesterol-conjugated form of resiquimod (R848) ProLNG-S, formulated as a positively charged liposome ProLNG-001, on murine BMDC and human PBMC. Furthermore, the manuscript evaluates the effects of the ProLNG-001 co-administered with ovalbumin (OVA) protein on the tumor growth of the melanoma in the B16-OVA model and compared the tumor progression and the survival of mice with the AS01 adjuvant co-administered with OVA protein. They found that ProLNG-001 induces the production of the proinflammatory cytokines by BMDC and human PBMC and that the ProLNG-001+OVA has an impact on tumor regression in the B16-OVA model. These are interesting finding, however there are some concerns that should be addressed.

1.       It is not clear from the data if the effect of the treatment on tumor growth and mice survival is due to the antigen specific immune responses or to the activation of the innate immune response due to the adjuvant effect. It would be useful include as control mice treated with the adjuvant (ProLNG-001) alone in the absence of OVA antigen. Indeed, the production of the pro-inflammatory cytokines by BMDC and PBMC is induced by the adjuvant alone. 

2.       The Authors should show the measurement of IgG Titer in Mouse Serum and the Analysis of Delayed Cytokine Response in Mouse Serum as they stated in the material and methods.

Minor points:

1.       The Authors should clarify if they use heparinized blood as they stated on line 157 page 4 or not heparinized. In the former case they should clarify that they used plasma instead serum.

2.       On line 192 page 5 is mentioned anti-CD3 twice.

Reviewer 3 Report

It is an interesting piece of work on cancer vaccine, by Kim et al. where they have developed and used a cholesterol-conjugated form of resiquimod (R848) called ProLNG-S to enhance immune efficacy while reducing toxicity. Formulated as ProLNG-001, a positively charged liposome, it was co-administered with ovalbumin (OVA) protein in the B16-OVA model. ProLNG-001 effectively targeted secondary lymphoid organs, resulting in a robust systemic anti-tumor immune response and activation of tumor-specific T cells.

The experiments were well designed, nicely executed and overall presentation is also fine.

I recommend acceptance in present form.

Fine

Author Response

We sincerely appreciate your positive feedback and insightful evaluation of our work on cancer vaccines using ProLNG-001. We are delighted to hear that you found our experiments to be well-designed and skillfully executed. Your recommendation for acceptance in its present form is both gratifying and motivating, and we are truly thankful for your support.

Reviewer 4 Report

In the Materials and Methods, line 84, what kind of filter used? PES, Nylon or PVDF?

In the Hematoxylin and Eosin Staining from which animals come from the tissues used? What kind of microscope used? What magnification?

In the paragraph 2.8, it is necessary to explain because choose this kind of tumor model. How many mice were used? How tumors were measured? ultrasound, RSM?

In the Discussion, you hypothesize the involvement of MYD88, but you have not gone to investigate this mechanism.

Round 2

Reviewer 1 Report

The manuscript by Kim et al addresses the effect of TLR7/8 agonist ProLNG001 as adjuvant for cancer treatment. This manuscript is presenting interesting and novel data and authors did improvements in the revised manuscript. Modulation of TLR/MyD88/TRIF signalling by ProLNG001 will provide essential data about therapeutic potential of this TLR7/8 agonist. This could be explored in the study by measurement of IRFs, IP-10 or interferons in response to ProLNG001 and could benefit the quality of the manuscript.

Reviewer 2 Report

The Author addressed the concerns and answered to the questions.

The manuscript is now more clear and complete.

Author Response

We sincerely appreciate your positive feedback and insightful evaluation of our work on cancer vaccines using ProLNG-001.

Thanks to your advice, we could develop our study and clarify the efficacy of ProLNG-001. Your recommendation for acceptance in its revised form is both gratifying and motivating, and we are truly thankful for your support.